# Interaction of plasticity and circuit organization during the acquisition of cerebellum-dependent motor learning

Yan Yang[1], Stephen G Lisberger[1,2]*

[1]Department of Neurobiology, Duke University School of Medicine, Durham, United States; [2]Howard Hughes Medical Institute, Duke University School of Medicine, Durham, United States

**Abstract** Motor learning occurs through interactions between the cerebellar circuit and cellular plasticity at different sites. Previous work has established plasticity in brain slices and suggested plausible sites of behavioral learning. We now reveal what actually happens in the cerebellum during short-term learning. We monitor the expression of plasticity in the simple-spike firing of cerebellar Purkinje cells during trial-over-trial learning in smooth pursuit eye movements of monkeys. Our findings imply that: 1) a single complex-spike response driven by one instruction for learning causes short-term plasticity in a Purkinje cell's mossy fiber/parallel-fiber input pathways; 2) complex-spike responses and simple-spike firing rate are correlated across the Purkinje cell population; and 3) simple-spike firing rate at the time of an instruction for learning modulates the probability of a complex-spike response, possibly through a disynaptic feedback pathway to the inferior olive. These mechanisms may participate in long-term motor learning.

*For correspondence:
lisberger@neuro.duke.edu

**Competing interests:** The authors declare that no competing interests exist.

**Reviewing editor**: Dora Angelaki, Baylor College of Medicine, United States

## Introduction

Learning is an adaptive change in behavior that results from 'plasticity' in the cellular mechanisms of synaptic transmission and/or spike generation in the nervous system. Our thinking about how the brain learns has been influenced heavily by the discovery of a form of synaptic plasticity called long-term potentiation (aka 'LTP'). Through research on reduced preparations, and electrical activation of neural pathways in artificial patterns, we have a remarkable catalog of multiple forms of cellular plasticity in neurons. But a fundamental bridge is missing. We need to monitor the plasticity that occurs in a known circuit when an animal is participating in a learning task, to reveal what actually happens in the brain during behavioral learning.

Our premise is that behavioral learning involves multiple forms of plasticity at multiple sites in an essential neural circuit for the behavior. Different forms of plasticity may be deployed at different sites over different time courses during learning. Circuit dynamics will play a key role in converting plasticity to learning. Thus, it is critical to understand when and where different forms of plasticity contribute to learning, and how circuits transform plasticity into changes in neural activity patterns. We start by analyzing a form of learning that occurs on the time scale of a single learning trial, with the premise that the analysis of short-term events will provide insight into circuit dynamics and plasticity that might play roles during long-term learning.

The cerebellum provides an excellent brain site to ask what happens in a neural circuit during learning. Many forms of cellular plasticity are present in the cerebellum (*Hansel et al., 2001*; *Zheng and Raman, 2010*; *Carey, 2011*; *Gao et al., 2012*). The cerebellar circuit is understood well, and there is a prominent theory of cerebellar motor learning (*Marr, 1969*; *Albus, 1971*; *Ito, 1972*) based on climbing-fiber induced long-term depression of the synapses from parallel fibers to Purkinje cells (*Ito and Kano, 1982*). Research

**eLife digest** Practice makes perfect in many areas of life, such as playing sport or even just drinking coffee from a cup without spilling any. Our brains can learn and improve these motor skills through trial, error and learning, with such "motor learning" depending on the cerebellum, a part of the brain that helps to coordinate all kinds of movements.

Motor learning is a product of the organization of the cerebellar circuit, which is well understood, and the "plasticity" in the synapses that determine how cerebellar neurons interact with each other. The cerebellum contains cells called Purkinje cells that receive distinctive inputs from two pathways: a pathway involving inputs from many parallel fibers, which convey signals related to sensory events or motor commands; and a pathway involving input from a single climbing-fiber, which conveys signals from a part of the brain called the inferior olive nucleus.

Research on slices of brain has revealed many sites and forms of cerebellar plasticity that could participate in motor learning. In one form of plasticity, the strength of the synapses between the parallel fibers and the Purkinje cell can be changed when a signal sent along the climbing fiber arrives the Purkinje cell.

Yang and Lisberger have now taken the next step by studying the cerebellum of a monkey as it performs a motor learning task. Remarkably these experiments show that the climbing fiber inputs cause plasticity of Purkinje cell activity, just as happens in the experiments on brain slices. Further, some learning in the cerebellum restricts further learning, so that the cerebellum puts boundaries on its own learning. Overall the results make clear how learning is a property of groups of neurons working together in a circuit, rather than simply of changes in the strength of specific synapses.

By shedding light on what happens in the cerebellum during short-term motor learning, the work of Yang and Lisberger will benefit efforts to understand how the cerebellum is involved in motor learning on all time scales.

in intact, learning animals using neural recordings, inactivation methods, and genetically-altered mice supports the cerebellar learning theory broadly (e.g., *Gilbert and Thach, 1977*; *Lisberger, 1994*; *Medina et al., 2000*; *Christian and Thompson, 2003*; *van Alphen and de Zeeuw, 2002*; *Boyden et al., 2004*; *Blazquez et al., 2006*; *Ke et al., 2009*; *Wulff et al., 2009*). As research on motor learning has advanced, we have come to understand that learning occurs at multiple neural sites and over several time scales (*Medina and Mauk, 2000*; *van Alphen and de Zeeuw, 2002*; *Jorntell and Ekerot, 2003*; *Boyden et al., 2004*; *Yang and Lisberger, 2010*).

In the present paper, we use learning in the smooth pursuit eye movements of awake, behaving monkeys to unravel how a neural circuit creates short-term cerebellar learning (*Kahlon and Lisberger, 1996, 2000*; *Medina et al., 2005*; *Medina and Lisberger, 2008*; *Yang and Lisberger, 2010*). The cerebellar circuit (*Figure 1*) includes a mossy-fiber/granule-cell/parallel-fiber pathway to Purkinje cells, and a climbing fiber pathway that originates in the inferior olive (IO). The parallel-fiber inputs cause conventional action potentials in Purkinje cells known as 'simple-spikes'. An action potential on the climbing-fiber input causes an unconventional, long-duration event known as a 'complex-spike' or 'CS'. Our formulation of the cerebellar circuit includes plasticity of Purkinje cell simple-spike responses driven by conjunction of parallel fiber and climbing fiber inputs (light pink shading in *Figure 1*), a disynaptic pathway from Purkinje cells to the inferior olive that includes two inhibitory synapses (light gray shading in *Figure 1*), and a disynaptic pathway from Purkinje cells to motoneurons.

We show how the circuit in *Figure 1* can explain a number of related observations on the spiking activity of Purkinje cells during short-term behavioral learning. First, we demonstrate that simple-spike firing rate predicts the probability of a complex-spike response to an instruction on a given trial, and we suggest that the disynaptic inhibitory pathway from Purkinje cells to the inferior olive causes this correlation (gray shading in *Figure 1*, *De Zeeuw et al., 1998*; *D'Angelo, 2010*; *Bengtsson and Hesslow, 2006*). Feedback from Purkinje cells might allow the cerebellum to control some of the error signals that guide synaptic plasticity in the cerebellum (*Mauk and Donegan, 1997*; *Kenyon et al., 1998*; *Miall et al., 1998*; *Bengtsson and Hesslow, 2006*). Second, we provide evidence that complex-spike-linked trial-over-trial depression of simple-spike responses (*Medina and Lisberger, 2008*) is an expression of short-term plasticity (pink shading in *Figure 1*), presumably in the mossy-fiber/parallel-fiber

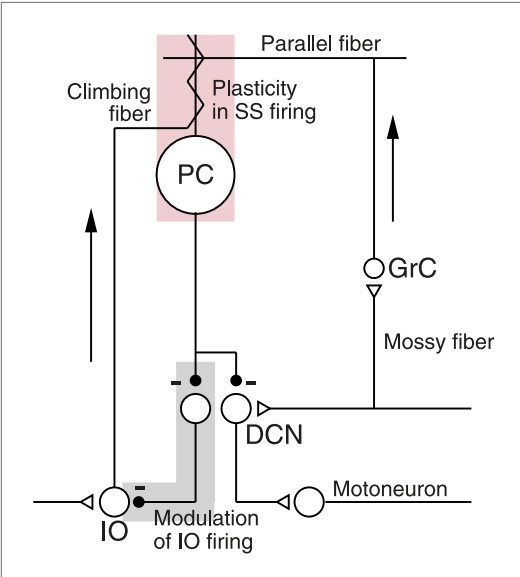

**Figure 1**. Schematic diagram of the cerebellar microcircuit. Abbreviations are: PC, Purkinje cell; GrC, granule cell; DCN, deep cerebellar nucleus; IO, Inferior olive. Pink shading highlights our finding of CS-linked plasticity in simple-spike firing. Gray shading highlights evidence that Purkinje cell output might regulate its own climbing-fiber inputs. Arrows indicate the flow of signals into the cerebellar cortex.

input pathways. Our paper suggests how plasticity interacts with a neural circuit during a form of short-term motor learning, and outlines circuit dynamics that also may play a role in long-term learning.

## Results

One of our previous papers (*Medina and Lisberger, 2008*) demonstrated that a complex-spike (CS) response on one learning trial leads to a 'trial-over-trial' depression of simple-spike responses on the subsequent trial. The depression is expressed selectively over a brief interval centered at the time when the instruction occurred. Another of our papers (*Yang and Lisberger, 2010*) showed that pursuit eye movements show 'trial-over-trial' learning when a monkey pursues a target motion with a change in direction. The eye velocity on one trial undergoes a transient and properly-timed deflection that takes the eyes in the direction of the change in target motion on the previous trial.

Our earlier findings, and the techniques they used, have set the stage for the approach deployed here. Our long-term goal is to understand what happens in the brain to cause motor learning when an instructive target motion is repeated many times. We start toward that goal in the present paper through analysis of events that occur after a single instructive stimulus. We put the approaches from our prior papers together in a way that (i) elucidates the multiple components, including CS-linked short-term plasticity, that cause trial-over-trial depression of simple-spike firing rate, and (ii) provides neural and behavioral evidence that the cerebellum may control its own instructive inputs in a way that would modulate cerebellar learning capacity.

### The learning paradigm

We use an experimental design that causes learned changes in the direction of smooth pursuit eye movements (*Medina et al., 2005*; *Yang and Lisberger, 2010*). In other words, visual targets start moving in one direction and, 250 ms later, undergo a change in direction. Pursuit learns to anticipate the change in target direction, leading to a small deviation in the direction of eye velocity after a single learning trial, and larger learned changes over blocks of 100 identical learning trials.

*Figure 2A* uses traces of eye position and velocity as a function of time to provide details about the structure of each learning target trajectory. Target motion began at a speed of 20 deg/s downward, in what we will call the 'pursuit' direction. As illustrated in the vertical position traces, the start of the downward ramp of target motion was accompanied by small upward displacement of the target to obviate the need for a catch-up saccade, thereby providing uncontaminated initiation of downward pursuit. Then, 250 ms after the onset of target motion, the target underwent an 'instructive' change in direction through the addition of a rightward or leftward ramp of motion at 30 deg/s for 400 ms (horizontal position traces in *Figure 2A*). The brief addition of horizontal target motion during downward target motion created learning target trajectories in a two-dimensional position space shown by the zigzags at the top of *Figure 2A*.

In most of the experiments reported here, we used a fixed pursuit direction, but delivered instructions that comprised changes in direction randomly in one of the two directions along the orthogonal, learning axis. This 'random-order' learning paradigm has the advantage that it allows us to study trial-over-trial learning while preventing any long-term learning. Trial-over-trial learning causes the eye movement on any given 'test' trial to possess a small, properly-timed deflection that takes eye velocity

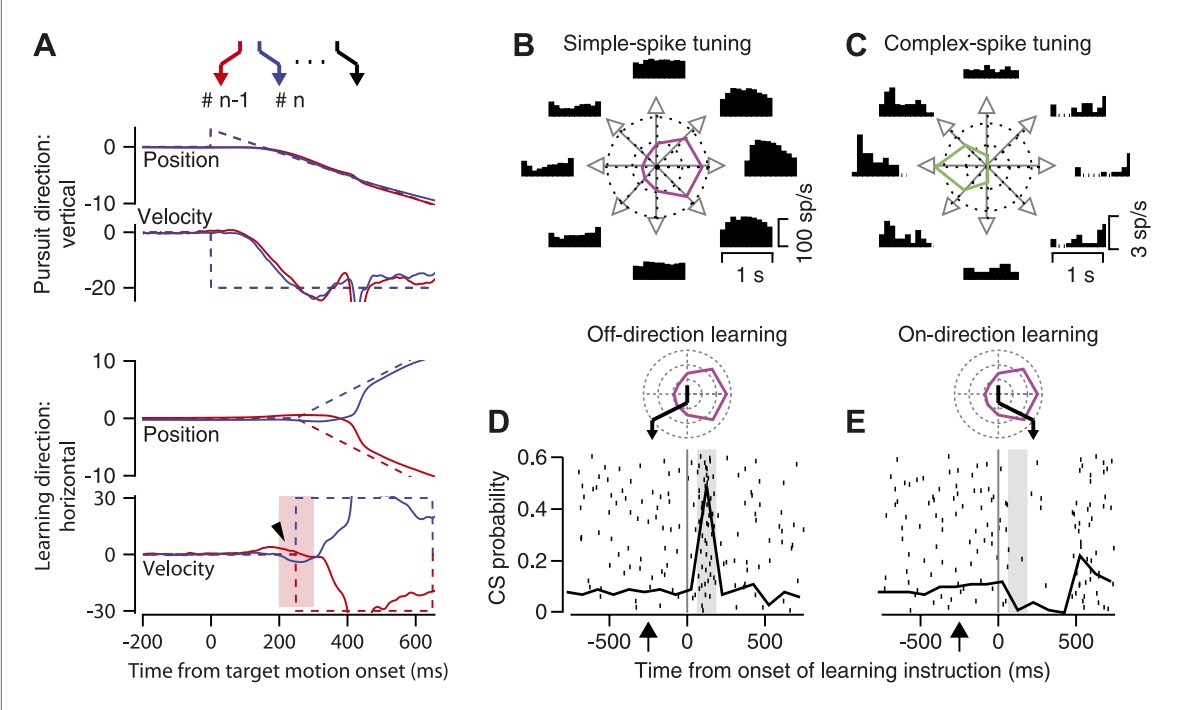

**Figure 2**. Background information about trial-over-trial learning and the responses of an example floccular Purkinje cell. (**A**) The zigzags at the top of the panel show a sequence of target motions in a random-order learning block. From top to bottom, the superimposed traces show vertical position, vertical velocity, horizontal position, and horizontal velocity as a function of time from the onset of target motion. Dashed and solid traces show target and eye movement. Different colored traces show responses in consecutive trials. The arrowhead on the horizontal velocity records points out trial-over-trial learning. The pink shading shows the analysis interval for trial-over-trial learning. (**B** and **C**) Direction-tuning of simple-spike (**B**) and complex-spike (**C**) responses of an example Purkinje cell. Histograms at eight locations show time varying firing rates for target motion in different directions. Polar plots show tuning-curves. (**D** and **E**) Superimposed rasters of CS responses and graphs of the probability of CS responses as a function of time from instruction onset for instructions in the off-direction (**D**) and on-direction (**E**) for simple-spike responses. The vertical line shows the time of the instruction onset and the gray shading shows the CS analysis interval. The polar plots above the CS rasters show the direction tuning of the example Purkinje cell in magenta, and use the black zigzag to indicate the trajectory of the learning target motions.

in the direction of the change in target motion on the prior 'instruction' trial. In the examples in *Figure 2A*, the small rightward deflection of horizontal eye velocity (arrowhead) in the red trace ('n-1' trial) was learned as the result of a rightward instruction in the prior trial ('n-2', not shown); the leftward deflection in the blue trace ('n' trial) was learned as the result of the leftward instruction on the 'n-1' trial. Because the eye velocity responses start before the instruction, they must be related to the instruction in the prior trial, rather than to that in the current trial.

We realize that trial-over-trial learning is not an adaptive response during a random-order learning block, because it causes an inappropriate learned response half of the time. We also realize that the trial-over-trial effects we report in the present paper probe only the initial phases of behavioral learning. Still, we regard the random-order paradigm as a means to get 'under-the-hood' and determine how neural circuits transform cellular changes into behavioral learning in a limited time frame. We expect that the phenomena we are studying also play a role in more natural learning conditions, when the same instructive stimulus occurs repeatedly.

## Response properties of floccular Purkinje cells

We recorded from Purkinje cells in the floccular complex of the cerebellum because prior research has demonstrated that they play important roles in both normal pursuit (***Miles and Fuller, 1975***; ***Lisberger and Fuchs, 1978***) and pursuit learning (***Kahlon and Lisberger, 2000***; ***Medina and Lisberger, 2008***). Under normal, pre-learning conditions, the simple-spike firing rate of floccular Purkinje cells is tuned for the direction of pursuit. For example, *Figure 2B* illustrates histograms of firing rate during pursuit in 8 directions, as well as the direction-tuning curve, for a Purkinje cell that responded best during pursuit

to the right. The CS responses of the same Purkinje cell were tuned for the opposite, leftward direction (*Figure 2C*), and had much lower firing probabilities. Opposite direction-tuning for simple-spike and CS responses is a general rule in the floccular complex.

For each Purkinje cell, we contrived a learning target trajectory that matched the cell's direction tuning. The initial target motion was orthogonal to the best direction for the simple-spike firing of the Purkinje cell under study, for example downward in *Figure 2*. The instructive change in target direction was in either the on-direction or the off-direction for the simple-spikes of the Purkinje cell, that is leftward or rightward in *Figure 2*. For instructions in the off-direction for simple-spike firing (*Figure 2D*), the CS responses occurred with a high probability in response to the instruction. For instructions in the on-direction for simple-spike firing (*Figure 2E*), the low spontaneous rate of CS responses was inhibited by the instruction.

## Selection of analysis intervals and a brief guide to the figures

In *Figure 2D,E*, and all subsequent graphs, time zero on the x-axis indicates the time of the instructive change in target direction. The onset of pursuit target motion always began 250 ms before the time of the instruction and is indicated by an upward arrow below each x-axis. We have indicated the analysis interval in each figure by shading, using gray shading to indicate the interval for analysis of the probability that an instruction evokes a CS, and light pink shading to indicate the analysis window for simple-spike firing rate.

We used the interval from 75 to 175 ms after the instruction (gray shading in *Figure 2D,E*) for the analysis of CS responses throughout the paper. Essentially all the CS responses to the instruction occurred in this interval. We used the interval from 50 ms before to 50 ms after the onset of the instruction (light pink shading in bottom panel of *Figure 2A*) to measure behavioral learning, and the simple-spike correlates of learning. This interval comprises the time when trial-over-trial effects were expressed in both neural responses and pursuit behavior. Importantly, the analysis interval ends before the instruction in the current trial could have any effect on eye movement or neural responses, and before there was any chance of a CS response to the instruction.

## Relationship between simple-spike firing rate and complex-spike probability

Our first observation suggests a causal link between simple-spike firing rate and the probability that a CS response occurs in response to an instruction. Simple-spike firing rate at the time of an instruction was ~5 spike/s higher on average depending on whether or not a CS occurred later in the trial in response to an off-direction instruction. The relationship between simple-spike and CS responses was restricted to the time interval surrounding the onset of the instruction (vertical dashed line in *Figure 3C,D*), suggesting that the relationship is not caused by low frequency fluctuations of attention or other top-down influences. Note that the increased simple-spike firing rate preceded the time of the CS response and therefore cannot be attributed to a trivial explanation such as bleed-through of the CS spike into the train of spikes used to calculate simple-spike firing rate.

Similar analysis of the eye velocity responses revealed a tiny difference of ~0.1 deg/s that was present before the instruction and could be seen only in high-gain records (*Figure 3A*). The difference in eye velocity probably was too small to cause the presence vs absence of a CS response to the instructive stimulus. Low-gain eye velocity records (*Figure 3B*) showed that the eye movement responses to the instruction were not related the presence vs absence of a CS response, a control that argues against the possibility that some trials lacked CS responses simply because the monkey was not attending or was being lazy.

*Figure 3C,D* show that the simple-spike firing rate of a Purkinje cell before an instruction is related to whether or not an off-direction instruction evokes a CS on that learning trial. Because the variation in simple-spike firing rate is present before the CS response, we suspected that simple-spike firing rate could causally modulate the probability of a CS. However, the analysis described so far stops short of assessing how strongly simple-spike firing affects CS probability, and therefore also does not give insight into the possible strength of the proposed modulation. To assess the strength of the relationship, we next used simple-spike firing as the independent variable and assessed its impact on the probability of a CS response for each Purkinje cell. To do so, we created distributions of simple-spike firing rate in each trial averaged across the analysis interval (light pink shading on traces in *Figure 3E,F*). Then, we trisected the distributions at the mean simple-spike firing rate plus/minus 0.44 standard deviations, grouped the

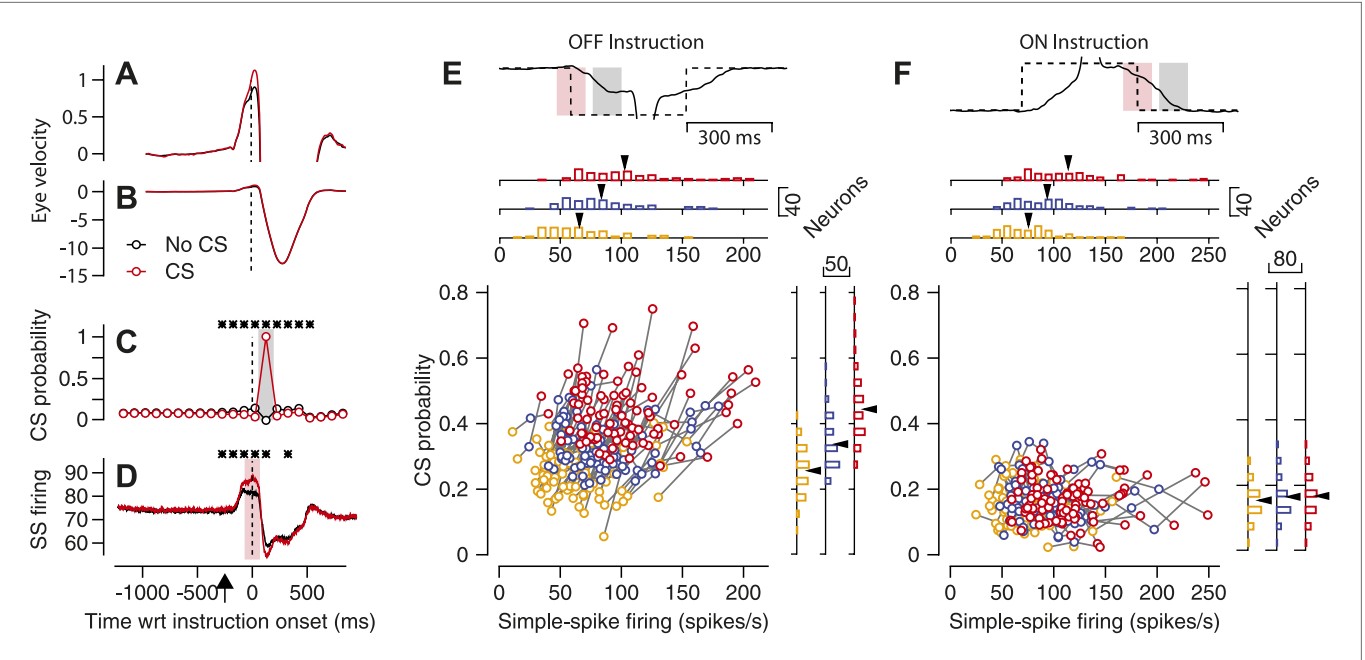

**Figure 3**. Relationship between simple-spike firing rate and probability of CS responses to an instruction. (**A**–**D**) Average eye velocity along the learning axis (**A** and **B**), CS probability (**C**), and simple-spike firing rate (**D**) as a function of time in off-direction learning trials. Red and black traces show averages for trials with and without a CS response to the instruction. The same data appear in (**A** and **B**), but with high and low gains on the eye velocity axis. Asterisks in (**C** and **D**) indicate 100-ms bins when the traces differed significantly (p<0.01, two-tailed paired *t*-test). Time is relative to the onset of the instruction and the upward arrow in **D** indicates the time of onset of target motion. (**E** and **F**) Neuron-by-neuron analysis for the start of an off-direction instruction (**E**) and the end of an on-direction instruction (**F**). The continuous and dashed traces at the top show eye and target velocity along the learning axis. The pink and gray shading show analysis intervals for simple-spike and CS responses. The scatter plot contains three symbols for each Purkinje cell (n = 90); yellow, blue, and red symbols show data for the lowest, middle, and highest third of simple-spike firing rates across trials for each Purkinje cell. Marginal histograms summarize the distributions across the population of Purkinje cells for the three different levels of simple-spike firing rate. Black arrowheads indicate the mean for each distribution.

trials for each third of the distribution, and computed the probability of a CS response to the instruction (analysis interval shown by light gray shading in *Figure 3E,F*) for each third of the distribution.

Each neuron showed a clear relationship between the probability of an instruction-evoked CS and the average simple-spike firing rate at the time of the instruction (*Figure 3E*). The clouds of points for different neurons move from lower left to upper right as we step from the trials with the lower 1/3 of simple-spike firing rates (yellow symbols) through the trials with the middle 1/3 (blue symbols) to the trials with the upper 1/3 (red symbols). The trend appears in the scatter plot showing three connected symbols for each of the 90 Purkinje cells in our sample, and in the marginal histograms. On average, the magnitude of simple-spike firing rate averaged 66, 84, and 103 spikes/s for the three segments of the distributions and the probability of a CS response averaged 0.26, 0.34, and 0.44. Correlation analysis of the cluster of points in *Figure 3E* yielded a statistically-significant correlation of 0.33 (p<<0.001). The slope of the means from the marginal histograms indicated that one spike/s of simple-spike firing rate is associated with a difference of 0.005 in the probability of a CS response to the instruction. We conclude that simple-spike firing rate may modulate the probability of CS responses, but weakly.

Another fact argues that simple-spike firing rate merely modulates CS probability and does not exert obligatory control. For the interval at the end of the pulse of instructive target motion in the on-direction for simple-spike responses, the relationship is quite weak between the probability of a CS and simple-spike firing rate at the time of the instruction (*Figure 3F*). The cluster of points in *Figure 3F* yielded a correlation of −0.028 that was not statistically significant (p=0.65) In this interval, stopping the target during on-direction eye motion causes image motion (relative to the retina) in the off-direction and evokes the CS responses. But this is not a strong learning situation, and that may explain the absence of a relationship between CS probability and simple-spike firing rate.

We have named the relationship between CS probability and simple-spike firing rate 'same-trial facilitation', to distinguish it from CS-linked 'trial-over-trial depression' (e.g., *Medina and Lisberger, 2008*). The existence of same-trial facilitation (*Miall et al., 1998*) raises the possibility we will consider in our 'Discussion', that same-trial facilitation reflects an important feature of cerebellar learning (*Mauk and Donegan, 1997*; *Kenyon et al., 1998*).

## Evidence for trial-over-trial plasticity in simple-spike firing

Our second observation shows that plasticity of simple-spike firing is linked to the occurrence of a CS response to an instruction. We showed in a prior publication that a CS response to an instruction on one trial is linked to a depression of simple-spike firing rate on the next trial (*Medina and Lisberger, 2008*). We demonstrated trial-over-trial depression there (and here) by dividing the stream of learning trials into pairs where the first trial of the pair delivered an off-direction instruction. In each pair, we called the first trial the 'instruction' trial, and the second trial a 'test' trial. We computed the millisecond-by-millisecond difference between the simple-spike firing rates on the two consecutive trials. We tagged the trial-over-trial difference in firing rate for each pair according to whether or not the instruction trial contained a CS response to the instruction. A '1-0' pair, for example, meant that the instruction trial contained a CS response to the instruction and the test trial did not. We then averaged the trial-over-trial changes in simple-spike firing rate, first for all like pairs of trials within a single Purkinje cell, and then across Purkinje cells.

Trial-over-trial depression in simple-spike firing rate appeared if the instruction trial contained a CS response to the instruction and the test trial did not (*Figure 4A*, red trace, '1-0' pairs); trial-over-trial

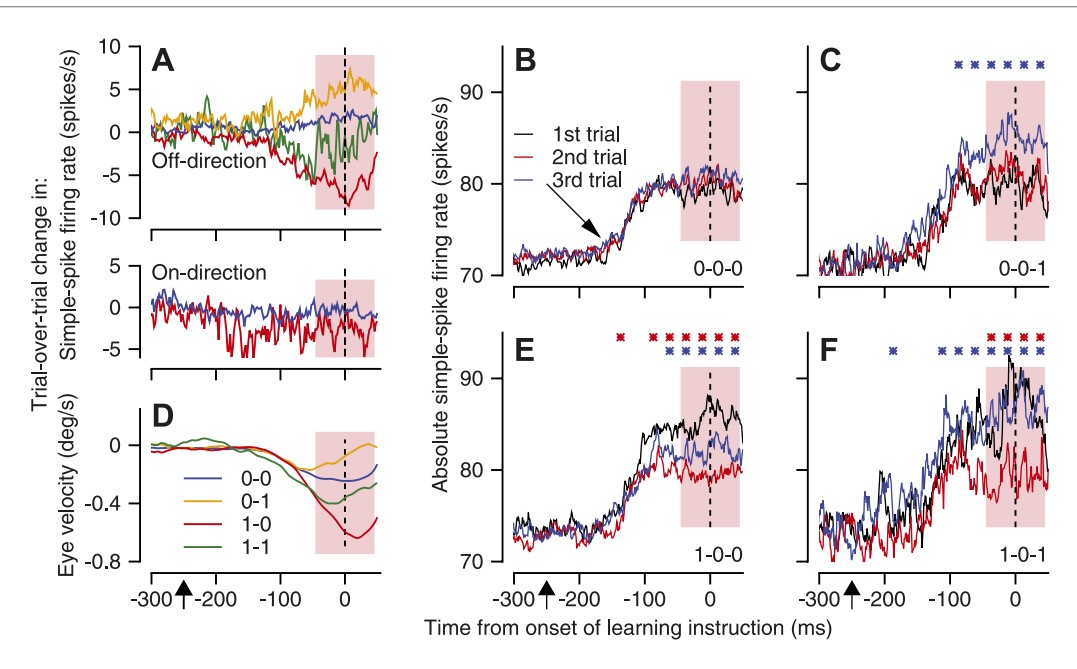

**Figure 4**. Mechanisms of CS-linked trial-over-trial depression of simple-spike firing rate and same-trial facilitation. (**A** and **D**) Trial-over-trial change in simple-spike firing rate and eye velocity. Different colors are for pairs of successive trials with different combinations of the presence or absence of a CS response to the instruction. In **A**, the top graph shows data for the onset of off-direction instructions and the bottom graph shows data for the offset of on-direction instructions. (**B**, **C**, **E** and **F**) Each graph plots the average simple-spike firing rate for trios of trials with different blends of the presence or absence of CS responses to an instructive change in the direction of target motion. Black, red, and blue traces show the first, second, and third trials of the trio. We performed statistical comparison in 25-ms bins using a paired two-sided Wilcoxon signed rank test. Red and blue asterisks indicate time bins when the firing rate in the second trial was statistically different from the first, or the third trial was statistically different from the second (p<0.05). Diagonal arrow in **B** points to a small increase in simple-spike firing rate related to the onset of pursuit target motion in a direction orthogonal to the preferred-null axis. Pink shading indicates the analysis interval. The x-axes show time with respect to the onset of the instruction. Upward arrows below the x-axes indicate the onset of pursuit target motion.

depression did not appear when neither trial contained a CS response (*Figure 4A*, blue trace, '0-0' pairs). It was tempting to assume that the depression was caused by CS-linked plasticity of simple-spike firing. However, the existence of 'same-trial facilitation' means that trial-over-trial depression could appear simply because of elevated simple-spike firing rate on the instruction trial, but not on the test trial. Thus, our next step is to take advantage of a dramatically larger dataset than we had before (*Medina and Lisberger, 2008*) to show that CS-linked short-term plasticity is one component of trial-over-trial depression.

We isolated a contribution of CS-linked plasticity by comparing the trial-over-trial depression of simple-spike firing for pairs of consecutive trials that both had CS responses ('1-1' pairs) with that for pairs of consecutive trials that both lacked CS responses ('0-0' pairs). Because the '1-1' and '0-0' pairs have the same CS response on each trial, they also should have the same amount of same-trial facilitation on each trial; same-trial facilitation would be 'nulled' and would not contribute to trial-over-trial depression. Even with the effect of same-trial facilitation nulled, the upper graph of *Figure 4A* shows a difference in trial-over-trial depression of about 5 spikes/s between '1-1' pairs (green trace) and '0-0' pairs (blue trace). We attribute this component of trial-over-trial-depression to CS-linked plasticity that occurs during trial-over-trial behavioral learning, because it persists after we control for same-trial facilitation.

Overall, *Figure 4A* shows a complicated progression of differences between the different-colored traces because it summarizes the combined effects of same-trial facilitation and CS-linked plasticity of simple-spike firing rate. We can see the effect of same-trial facilitation by comparing pairs of trials that had the same CS response on the instruction trial, to hold trial-over-trial plasticity constant, but different CS responses on the test trial. For example, the simple-spike firing rate was the same in both trials of the pair when neither included a CS response ('0-0'), so the trial-over-trial change in firing rate was almost flat (*Figure 4A*, blue trace). When only the test trial included a CS response ('0-1', yellow trace), simple-spike firing rate was higher in the test trial and the trial-over-trial change in simple-spike firing rate was more positive. The same effect appears when a CS response always occurred in the instruction trial. The simple-spike firing rate was higher in test trials that contained a CS response (*Figure 4A*, '1-1', green trace), and the trial-over-trial change in firing rate was more positive, compared to pairs that did not contain a CS response in the test trial (*Figure 4A*, '1-0', red trace).

Finally, trial-over-trial learning in eye velocity (*Figure 4D*) is closely aligned to the trial-over-trial changes in simple-spike firing rate (*Figure 4A*) for the 4 pairs of trials, supporting a causal link between trial-over-trial depression in neural firing and trial-over-trial learning in behavior. As we have noted elsewhere (*Medina and Lisberger, 2008*), the fact that the learned eye velocities track the trial-over-trial changes in simple-spike firing rate implies that the CS responses are highly correlated across the Purkinje cell population. If the CS responses occurred with independent probabilities in different Purkinje cells, then the average across the population of Purkinje cells would be independent of the pattern of CS responses in the different pairs of trials. The learned eye velocity would be present for all pairs, and would not differ according to the pattern of CS responses.

To perform statistical testing, we averaged firing rate across the analysis interval for simple-spike firing rate in individual neurons. Comparison of different pairs revealed statistically significant differences among the four types of pairs of trials (multiple comparisons in a two-way ANOVA, n = 90 neurons, simple-spike firing, p<0.01; eye velocity, p<0.05).

Additional evidence for the existence of CS-linked plasticity during learning comes from analysis of trial-over-trial changes in simple-spike firing rate at the offset of on-direction instructive target motion. Same-trial facilitation is very weak at the offset of on-direction instructions (*Figure 3F*). However, comparison of the trial-over-trial changes in firing rate for '1-0' pairs and '0-0' pairs revealed about 5 spikes/s of CS-linked trial-over-trial depression in simple-spike firing rate (bottom graph in *Figure 4A*, also *Medina and Lisberger, 2008*). The difference in the firing rate averaged across the simple-spike analysis interval was statistically-significant (paired *t*-test, p=0.002, n = 90). The presence of trial-over-trial depression in conditions that lack same-trial facilitation, albeit with a slightly different time course, implies that trial-over-trial depression is caused partly by plasticity of simple-spike firing that is linked to CS responses.

## Cause of same-trial facilitation

Our third observation elucidates the components that contribute to CS-linked same-trial facilitation. We considered trios of trials instead of pairs, and evaluated how simple-spike firing rate evolved from

the first to the second to the third trial. This approach gives us better control of the history of CS responses and allows more definitive conclusions about the components of trial-over-trial changes in neural responses.

Our strategy was to divide trios of consecutive trials into groups defined by binary codes that indicate whether or not a CS response occurred in response to the instruction in each trial. We then computed the time-varying average simple-spike firing rate for trios with different patterns of CS responses to off-direction instructions. We required that the first two trials of each trio deliver an off-direction instruction, and we compared the simple-spike firing rate on the third trial after two trials with identical histories of CS responses: '0-0-0' vs '0-0-1'; and '1-0-0' vs '1-0-1'.

As expected, simple-spike firing rate was the same in all three trials of '0-0-0' trios (*Figure 4B*), suggesting that our strategy of isolating trios was successful at controlling the effects of past history. Each trial showed the same modest increase in simple-spike firing rate (diagonal arrow in *Figure 4B*) in response to the initial eye movements along the pursuit axis chosen for the Purkinje cell under study, but the firing rate in the analysis interval (pink shading) was the same for all three trials. However, for '0-0-1' trios, the simple-spike firing rate was higher in the third trial (*Figure 4C*, blue trace) as expected from the same-trial facilitation documented in *Figure 3*, given the presence of CS response to the instruction. The same result emerges from comparison of the responses in the third trial of '1-0-0' and '1-0-1' trios in *Figure 4E,F* (blue traces), with the caveat that the traces are noisier in *Figure 4F* because of a paucity of '1-0-1' trios.

For the comparisons made in the previous paragraph, the only difference in the stimulus conditions was whether or not the instruction caused a CS response in the third trial. Because the history was the same for each comparison, the most plausible explanation for the intricacies revealed by *Figure 4B,C,E,F* is that same-trial facilitation of simple-spike firing rate is related to spontaneous fluctuations in simple-spike firing rate. The same outside influence might drive simple-spike firing rate and the probability of a CS response in parallel. Or, higher simple-spike firing rates might operate through disynaptic disinhibition (e.g., *Figure 1*) to cause a higher probability of inferior olive (and therefore CS) responses to a given sensory event.

## Computational analysis of cerebellar operation during trial-over-trial depression and learning

Our data suggest that a number of features of cerebellar organization and function collaborate to create the specific fingerprint of cerebellar learning revealed in *Figures 3 and 4*. However, the most stringent test for the logic of our conclusions comes from analysis of a computational model that includes the features we think are important. The data we strove to mimic with the model were: the trial-over-trial changes in simple-spike firing rate and eye velocity for all pairs of trials with and without the CS responses to an instructive stimulus (*Figure 4A,D*); the absolute firing patterns for the relevant trios of trials (*Figure 4B,C,E,F*); and the relationship between CS probability and simple-spike firing rate (*Figure 3*).

We created a population of 1000 Purkinje cells and 100 inferior olive neurons and connected them according to the architecture outlined by the cartoon in *Figure 5*. The cartoon retains the two inhibitory synapses between Purkinje cells and inferior olive neurons to emphasize the sign of the putative pathway in the brain, even though the neuron in the deep cerebellar nucleus was not actually modeled (Equations appear in the 'Materials and Methods'). The model incorporated a number of features that were designed to allow it to reproduce our data, and that were suggested by data from our research and others':

- The simple-spike responses of each Purkinje cell varied from trial-to-trial. Each individual response, meant to reflect the simple-spike firing rate averaged across the analysis interval used for our data, was drawn from a distribution with a mean of 100 spikes/s and a standard deviation of 18 spikes/s, in agreement with our recordings (data not shown).
- The trial-to-trial variation in simple-spike firing rate was correlated at a level of ~0.16 between pairs of model Purkinje cells. This level of neuron-neuron correlation enables the neuron-behavior correlations found 250 ms after the onset of pursuit by *Medina and Lisberger (2007)*.
- Each inferior olive neuron provided a climbing fiber to 10 Purkinje cells (*Sugihara et al., 2004* visualized at least 5).
- The probability that an off-direction change in target motion will evoke a response in a particular inferior olive neuron on a given trial is modulated by the simple-spike firing rate of floccular Purkinje cells on that trial. On-direction instructions never evoke the CS responses in model Purkinje cells.

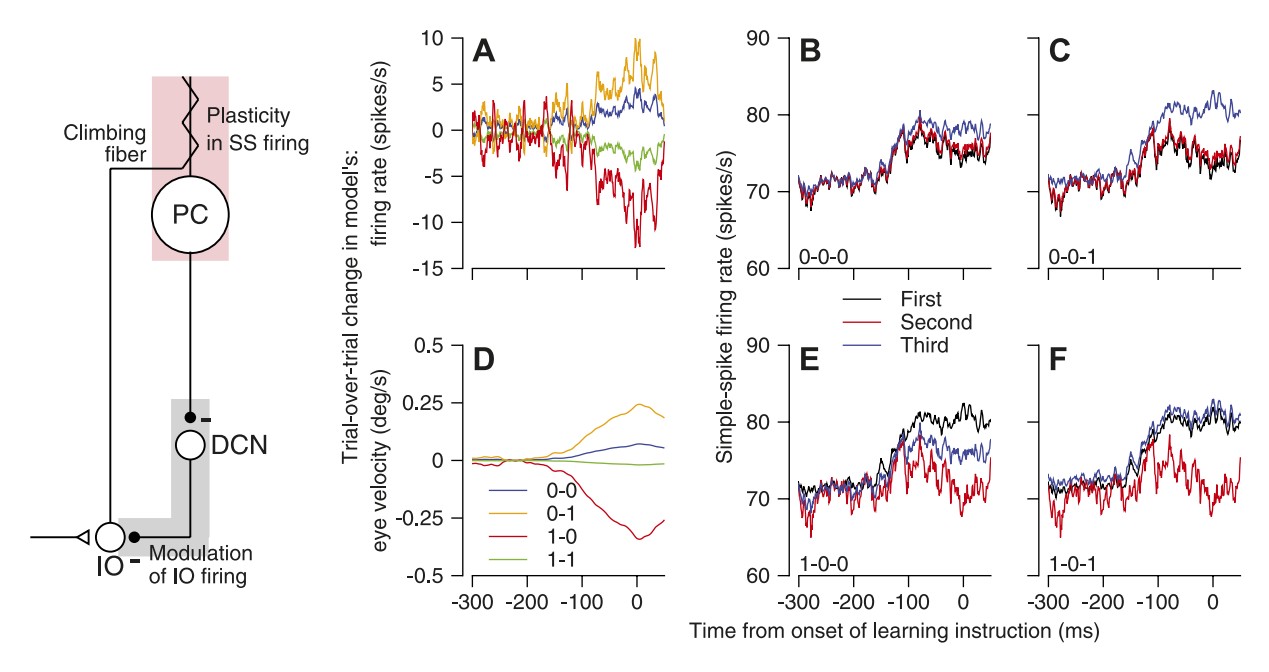

**Figure 5**. A cerebellar circuit model that reproduces the features of trial-over-trial depression of simple-spike firing rate and trial-over-trial learning in eye velocity. The schematic on the left shows the cerebellar circuit on which our model was based. Abbreviations are: PC, Purkinje cell; DCN, deep cerebellar nucleus; IO, inferior olive. (**A** and **D**) Trial-over-trial change in simple-spike firing rate and eye velocity in the model, after **Figure 4A,D**. (**B**, **C**, **E** and **F**) Predictions for the firing rate responses in the first, second, and third trials in trios with different combinations of CS responses, after the same panels in **Figure 4**. The x-axes show time with respect to the onset of the instruction.

- If a model Purkinje cell receives an input from its climbing fiber on a given trial, its simple-spike firing rate on the subsequent trial is reduced by 5 spikes/s through CS-linked plasticity.
- CS-linked plasticity relaxes back to zero over 2 trials when the instruction-test interval is 2.5 s (Yang and Lisberger, unpublished data).
- The responses of model inferior olive neurons were synchronized enough so that the trial-by-trial correlation in the occurrence of a CS between model Purkinje cells was ~0.25 on trials that delivered an off-direction instruction.

We ran the model for 800 trials, each of which delivered a randomly selected on-direction or off-direction instruction. We analyzed the 'data' from the model using the same approach we had used for the real Purkinje cells. We assumed that the eye movement on each trial was defined by the average of the responses of all 1000 Purkinje cells on that trial, scaled arbitrarily so that the full model produced eye velocities of the amplitudes in our data. Because we did not simulate the responses in each trial as a function of time, the outputs from the model are scalars on each trial. For demonstration purposes, we have used those scalars as gains on the relevant simple-spike firing rate and eye velocity traces from our data in **Figure 4**.

Analysis of the 'data' produced by the model yielded results that captured most of the features of our recordings from monkeys. The trial-over-trial changes in simple-spike firing rate (**Figure 5A**) and eye velocity (**Figure 5D**) showed the same progression as the data in **Figure 4A,D**. Simple-spike firing rate mimicked **Figure 4A** almost perfectly, but there are minor differences in polarity of the eye velocities for '0-1' and '0-0' pairs. We suspect that these differences are related to correlations in the CS responses across the population of Purkinje cells that we have not captured fully in our model. **Figure 5B,C,E,F** shows excellent agreement between the performance of the model and the data in **Figure 4B,C,E,F**. The model also reproduced our finding that each spike/s of simple-spike firing rate is associated with an increase of ~0.005 in the probability of a CS response to an off-direction instruction (data not shown).

Analysis of the sensitivity of the model to its parameters revealed that almost all of the features of the model were necessary to reproduce the data. In **Figure 6**, we have plotted the scalar results of

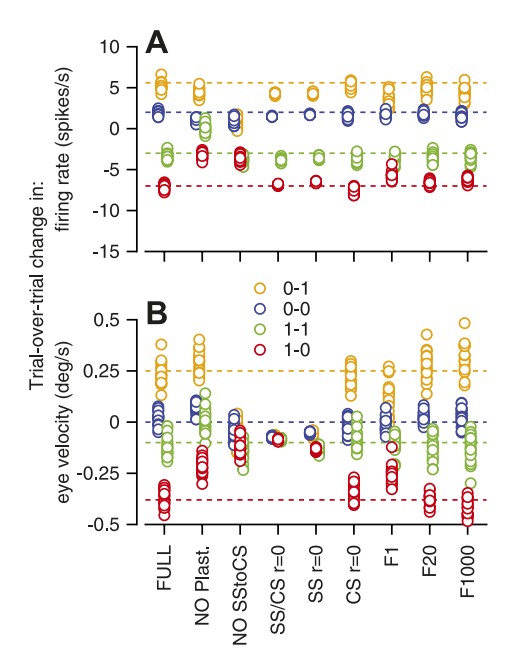

**Figure 6**. Sensitivity of the computer model to removal of different features. Different symbols show the amplitude predicted by the 20 runs of a given architecture of the model; different colors indicate different combinations of CS responses in successive instruction and test trials. Abbreviations for different architectures are: FULL, full model; NO plast, no trial-over-trial plasticity; NO SStoCS, no connection from DCN to IO; SS/CS r = 0; no neuron–neuron correlations in simple-spike or CS responses; SS r = 0; no neuron–neuron correlations in simple-spike responses; CS r = 0, no neuron–neuron correlations in CS responses; F1, F20, F1000, IO neurons receive inputs from 1, 20, or 1000 model Purkinje cells.

each of 20 runs of the model, with different configurations shown at different locations along the x-axis. The predictions for trial-over-trial changes in simple-spike firing rate broke down if we eliminated either CS-linked plasticity ('NO plast') or same-trial facilitation ('NO SStoCS'). Trial-over-trial depression of simple-spike firing rate was fairly normal for all other configurations of the model, and also did not depend strongly on whether the response of an inferior olive neuron to the instruction was controlled by 1, 20, or all 1000 model Purkinje cells ('F1', 'F20', 'F1000'). Trial-over-trial changes in eye velocity were quite sensitive to the presence of neuron–neuron correlations in the simple-spike firing rate and the CS responses across the population of Purkinje cells. Thus, the model supports the intuition outlined earlier about the likely high-degree of correlation in the presence and absence of CS responses across the population of Purkinje cells.

We conclude that the features included in our model are at least sufficient to reproduce the trial-over-trial changes in simple-spike firing rate and eye velocity in our data. Even though the features of the model are supported by knowledge of the organization of the cerebellar circuit, it is possible that other, quite different, models would make similar predictions. We also note that the model does not reproduce the absence of same-trial facilitation at the end of an on-direction learning stimulus, when CS-linked plasticity persists. Clearly, we do not understand completely the basis for CS-related same-trial facilitation.

## Testing predictions of the postulated pathway from Purkinje cells to the inferior olive

The microcircuit in *Figure 1*, and its success in modeling our observations on simple-spike firing rate (*Figure 5*), supports the hypothesis that a learned decrease in simple-spike firing rate may cause a decrease in the probability of CS responses to the instruction, and thereby reduce the magnitude or speed of learning. It could do so through the pathway from Purkinje cells through the deep cerebellar nucleus to the inferior olive. This pathway contains two inhibitory synapses so that increases or decreases in simple-spike firing rate would facilitate or inhibit responses of neurons in the inferior olive. The possibility that such a pathway exists and plays a functional role makes three testable predictions.

### Prediction #1

For a random-order learning block, a CS in response to an off-direction instruction on one trial will cause trial-over-trial depression of simple-spike firing rate on the next trial. If the second trial of a pair delivers an off-direction instruction, then the probability of a CS on that trial should be reduced. We tested this prediction by finding all pairs of trials that delivered successive off-direction instructions and asking whether the probability of '1-1' pairs was lower than predicted by the independent probability of a CS response to an off-direction instruction. For the full sample of 90 Purkinje cells, the probability of '1-1' pairs was 0.11 and the probability predicted by independent occurrence of CS responses was 0.12. Even though the difference between the actual and independent probabilities was only 0.01,

it was highly significant (two tailed paired *t*-test, p<<0.01). There was a commensurate slight excess in the probability of '1-0' pairs. The actual probability of 0.24 was significantly higher than the probability of 0.22 predicted by independence (two tailed paired *t*-test, p<0.01). The model in *Figure 5* showed the same small effects.

## Prediction #2

For a learning block where the same instruction occurs on multiple repeated trials, a gradual learned decrease in simple-spike firing should be associated with a gradual decrease in the probability that a CS occurs in response to an off-direction instruction. *Figure 5B* of *Medina and Lisberger (2008)* verifies this prediction, and our data agree with theirs (not shown).

## Prediction #3

Past behavioral learning should inhibit future behavioral learning. After some number of trials that deliver an off-direction instruction, both eye movements and simple-spike firing rate would learn. The neural learning would be expressed as a transient decrease in simple-spike firing rate surrounding the time of the instruction. Lower simple-spike firing should cause a reduction in the probability of CS responses to the instruction (see 'Prediction #2'). The reduced probability of CS responses should partially inhibit future learning. In the next paragraph, we describe a behavioral experiment that is consistent with the prediction that some past learning inhibits future learning.

To induce some learning but leave headroom for future learning, we delivered a series of learning trials that repeated the same direction of instruction, but with an instruction-test interval of 6 s instead of the usual interval of 2.5 s. The longer instruction-test interval supports less long-term and trial-over-trial learning than does a shorter instruction-test interval (Yang and Lisberger, unpublished data). Thus, in *Figure 7A,B*, the learning in the first 50 trials is lower in amplitude for the 6 s instruction-test interval (open blue symbols) than for the 2.5 s instruction-test interval (open red symbols). After the 50th learning trial, we switched to an instruction-test interval of 2.5 s. The learning curve (*Figure 7A,B*, filled blue symbols) initiated an increase in learning after the switch from long to short instruction-test intervals, but the magnitude of learning never caught up to the learning obtained when we used the short instruction-test interval for the entire learning block (filled red symbols). We assessed the effect of past learning on future learning by measuring the size of the learned eye velocity in the last 10 trials of each 100-trial block (gray shading in *Figure 7A,B*). The effect of past learning was instantiated by as 10 trials of learning, and became slightly stronger as a function of the number of trials that delivered the longer instruction-test interval in the early phase of learning blocks (*Figure 7C,D*). We conclude that some past learning inhibits future learning, as predicted if a learned depression of Purkinje cell simple-spike responses inhibits instructive climbing-fiber inputs.

## Discussion

In vitro studies have identified plasticity at multiple sites in the cerebellar cortex and the deep cerebellar nucleus, and have outlined many intricacies of the cerebellar circuit. The studies of neural responses in intact animals have measured neural representations of motor memories and have inferred what happened in the cerebellum during motor learning. There are numerous examples where genetic ablation of specific forms of cellular plasticity cause altered-learning phenotypes. Still, prior studies have stopped short of 'monitoring' what happens in the cerebellum during behavioral learning.

We have made the next step of creating a quantitative description of how CS-linked plasticity of simple-spike firing interacts with the cerebellar circuit in a behaving monkey to create motor learning. By combining the approaches from two of our prior studies (*Medina and Lisberger, 2008*; *Yang and Lisberger, 2010*) we have made four observations that relate to cerebellar operation during short-term motor learning:

1. A single CS causes short-term CS-linked plasticity of simple-spike firing during motor learning, perhaps due to synaptic depression at the parallel fiber to Purkinje cell synapse.
2. Higher simple-spike firing rate at the time of an instructive stimulus for learning is correlated with, and might cause, a higher probability of a CS response.
3. Learned depression of simple-spike responses is associated with, and might cause, a reduced probability of a CS response to an instruction.
4. Some past learning inhibits future learning.

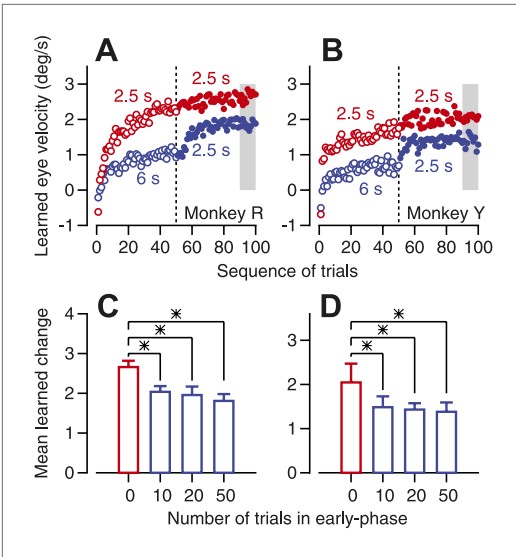

**Figure 7**. Evidence that past learning inhibits future learning. (**A** and **B**) Eye velocity learning curves for two monkeys. Each symbol shows the average learned eye velocity on a given trial in a sequence of 100 learning trials, averaged across 10 repetitions of the experiment. Red and blue open symbols show data for instruction-test intervals of 2.5 and 6 s in the first 50 trials of the block. Red and blue filled symbols show data for an instruction-test interval of 2.5 s in the second 50 trials of the block after an instruction-test interval of 2.5 and 6 s in the first 50 trials. (**C** and **D**) Averages of mean learned eye velocity at the end of a block of 100 learning trials as a function of the number of trials in the early phase with a long instruction-test interval. Asterisks indicate statistically-significant differences (two tailed *t*-test, p<0.01). The gray shading in **A** and **B** indicates the analysis window used for the graphs in **C** and **D**.

The model in *Figures 5 and 6* reproduced most of the features of our data by simulating CS-linked short-term plasticity of simple-spike firing, a disynaptic, disinhibitory pathway from Purkinje cells to the inferior olive, neuron–neuron correlations in simple-spike firing rate, and correlations in the occurrence of CS responses across the population of Purkinje cells. The success of the model shows that these four features of cerebellar anatomy and physiology are sufficient to create the physiological observations we have detailed in the present paper. The necessity of including all four features shows how plasticity and circuit organization could work together to create a complex but quantifiable set of relationships among different neural and behavioral phenomena.

We think our model might reflect the actual operation of the cerebellar learning system, because available knowledge supports the existence of the four critical features of the model. CS-linked trial-over-trial depression of simple-spike firing was demonstrated before (*Medina and Lisberger, 2008*), and we have verified here that trial-over-trial depression results partly from single-trial plasticity. The present paper demonstrates CS-related same-trial facilitation of simple-spike firing rate during learning, and *Miall et al. (1998)* showed the same phenomenon during quiescence. *Medina and Lisberger (2007)* provided evidence for moderate neuron–neuron correlations in simple-spike firing rate within the population of floccular Purkinje cells, and a number of papers have outlined the case for synchrony across the climbing-fiber inputs (*Llinas et al., 1974*; *Welsh et al., 1995*). Thus, our measurements have created a hypothesis for the operation of the cerebellar microcircuit during behavioral learning that is validated by experimental data and computer simulations. Still, we are aware that other models also could reproduce our data. For example, the probability of a CS and the simple-spike firing rate at the time of the instruction could be subject to top-down influences in parallel, a situation that would not require a pathway from Purkinje cells to the inferior olive, and might alter the implications for what happens in the cerebellum during motor learning.

The relationship between simple-spike firing rate on a given trial and the probability of a CS response to an instructive stimulus on the same trial has multiple implications. One implication is caution in assuming that trial-over-trial depression is due to CS-linked plasticity. Indeed, our analysis indicates that CS-linked plasticity is responsible for only half of trial-over-trial depression. The other half of trial-over-trial depression on '1-0' pairs of trials is related to the tendency for simple-spike firing rate to be higher on trials with a CS. A second implication concerns a possible role in learning. If simple-spike firing rate acts through inhibitory neurons in the deep cerebellar nucleus to modulate the response of inferior olive neurons to a given instruction, then a learned response in the cerebellum could modulate its own instructive inputs (*Schweighofer, et al., 2013*). There are two inhibitory synapses in the pathway from Purkinje cells to the inferior olive (*De Zeeuw et al., 1998*; *D'Angelo, 2010*), so higher (lower) simple-spike firing could cause a higher (lower) probability of CS responses, in agreement with the data of *Miall et al. (1998)* during quiescence and with our data during learning. Importantly, the climbing fiber axons from the inferior olive to the cerebellum and the pathway from the deep cerebellar nucleus back to the inferior olive both cross the midline, so that Purkinje cells can control their own climbing-fiber inputs (*Ruigrok, 1997*).

Three features of our data support the idea that learned changes in the responses of Purkinje cells could modulate their own instructive climbing fiber inputs (*Mauk and Donegan, 1997*; *Kenyon et al., 1998*; but see *Zbarska, et al., 2008*). (1) There is a correlation between CS probability and simple-spike firing rate on learning trials. (2) Learned decreases in simple-spike firing are associated with a reduction in the probability of CS responses to an instruction, both during learning blocks that repeat the same instruction on many trials (*Medina and Lisberger, 2008*) and during single-trial learning. (3) Past learning inhibits future learning, presumably by causing a learned decrease in the simple-spike responses of Purkinje cells (*Medina and Lisberger, 2008*) that could reduce the probability of the CS responses that would instruct learning. In addition, anatomical and physiological studies have documented the required neural pathway (*Ruigrok, 1997*; *Bengtsson and Hesslow, 2006*).

We suggest that the cerebellar learning circuit is set up to be conservative by allowing only a finite amount of learning in the cerebellar cortex. Learning may need to be transferred to the deep cerebellar nucleus (*Ohyama et al., 2006*) so that plasticity can relax back to baseline in the cerebellar cortex before further plasticity is allowed in the input pathways to Purkinje cells. Still, we are aware that there are multiple possible explanations for the observation that past learning inhibits future learning in pursuit eye movements. Thus, our data add support for, but do not obligate the conclusion that Purkinje cells modulate their own climbing-fiber inputs.

We have used physiological techniques to create a hypothesis (*Figure 8*) that starts to outline how the cerebellar circuit architecture and plasticity might interact during motor learning. Consider a sequence of trials that delivers the same instruction on 50 consecutive trials. A CS response on trial '1' leads to a properly-timed depression of simple-spike firing on trial '2'. Because of the decreased simple-spike firing, there is a lower probability of a CS response on trial '2'. On trial '3', the random variation in simple-spike firing rate leads to some trials with the same or higher firing than trial '1' and a higher probability of a CS response to the instruction; other trials have lower simple-spike firing rate and a lower probability of a CS response. Finally, our analysis predicts that by trial '50', sustained learning has depressed simple-spike firing rate substantially (*Medina and Lisberger, 2008*), so that the probability of a CS response is below control values and further learning is limited in the cerebellar cortex. Additional learning might become possible at a later time if some of the changes in the cerebellar cortex are transferred to the deep cerebellar nucleus, allowing a relaxation of the depression in the cerebellar cortex.

We expect that the CS-linked plasticity of simple-spike responses in our data is a product of CS-driven depression of the synapses from parallel fibers to Purkinje cells. For example, cannabinoid-dependent synaptic depression seems to have the requisite brief time course (*Brown et al., 2003*; *Brenowitz and Regehr, 2005*). However, other forms of cellular plasticity may contribute, either within the cerebellar cortex (*Hansel et al., 2001*; *Carey, 2011*) or in the deep cerebellar nucleus (*Miles and Lisberger, 1981*; *Zhang and Linden, 2006*; *Zheng and Raman, 2010*). The oculomotor vermis also could play a role in pursuit learning (*Suzuki and Keller, 1988*; *Takagi et al., 2000*). We expect that future work will provide positive evidence of specific roles during learning for other forms of plasticity and other brain sites of learning. Indeed, we hope that we have outlined an approach that might be used to establish roles for other forms of plasticity in the cerebellar circuit.

## Materials and methods

### Animal preparation

We conducted experiments on four awake, behaving adult male rhesus monkeys. Two monkeys were used for recordings from Purkinje cells in the floccular complex during pursuit learning (at UCSF), and the other two were used only for behavioral studies of pursuit learning (at Duke). During each experiment session, head movement was prevented with an implanted head holder, eye position was monitored with a scleral search coil, and a stainless steel recording cylinder allowed access to the floccular complex for single neuron recordings. The head holder, eye coil, and recording cylinder had been implanted in surgical procedures that used sterile technique with the monkey under isofluorane anesthesia (*Ramachandran and Lisberger, 2005*). Monkeys received analgesics for several days after each surgery. The Institutional Animal Care and Use Committees at UCSF and Duke had approved all the procedures in advance. The procedures were in accordance with the National Institutes of Health Guide for the Care and Use of Laboratory Animals.

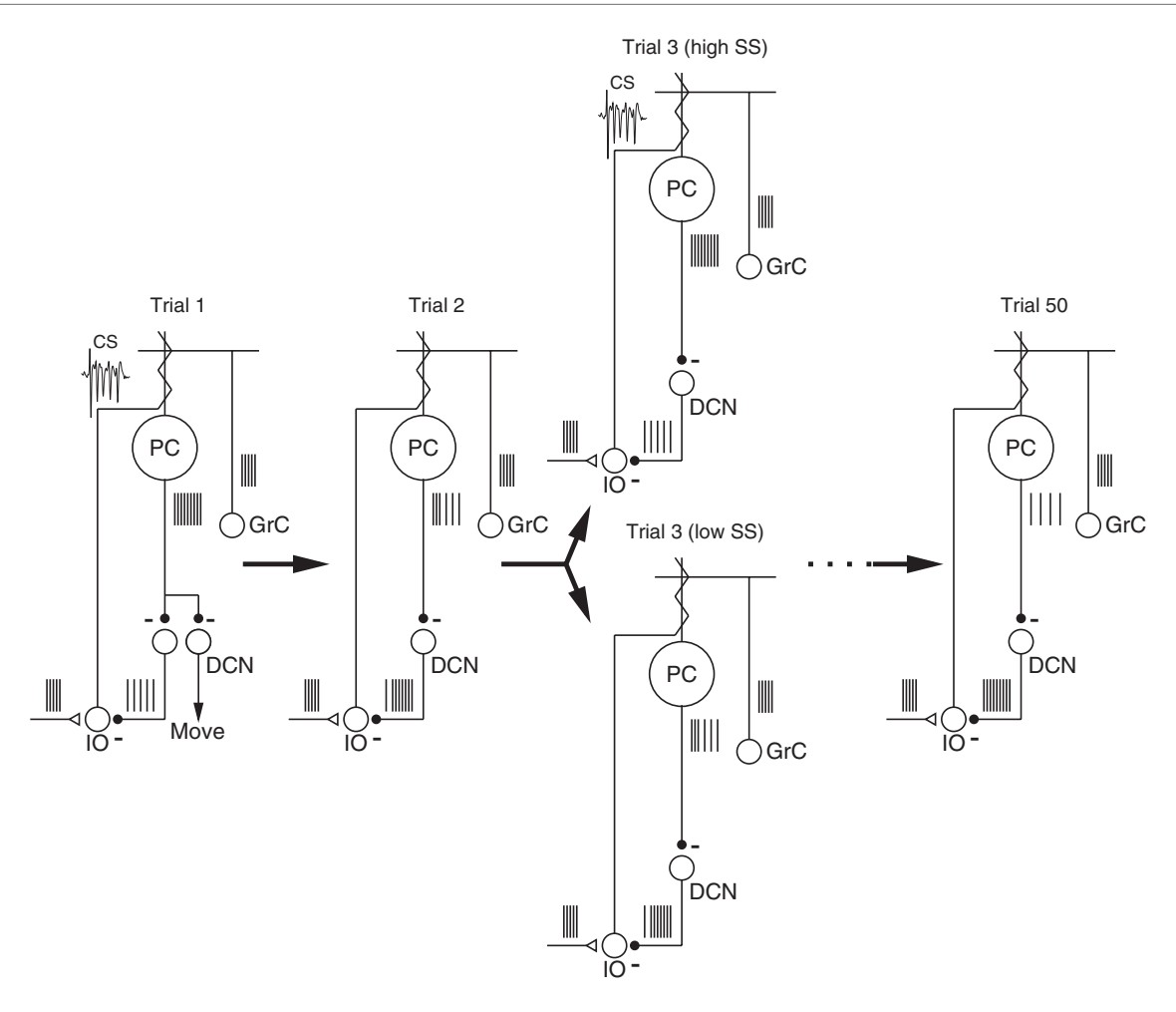

**Figure 8**. Schematic diagrams showing mechanisms of cerebellar learning suggested by our data. Each panel shows the spiking activity of the elements in the proposed cerebellar circuit during the course of a learning experiment. Trial '3' contains two schematics illustrating the effect of spontaneous trial-to-trial variation in the simple-spike firing rate, superimposed on the effects of trial-over-trial depression and longer-term learning. Abbreviations are: PC, Purkinje cell; GrC, granule cell; DCN, deep cerebellar nucleus; IO, inferior olive; CS, complex-spike.

## Behavioral task

Fixation and pursuit targets comprised bright 0.3° or 0.5° spots, respectively, on a dark background. Visual stimuli were presented on a CRT monitor that was placed 30 cm from the monkey's eye and that subtended a visual field of 59° × 47°. All experiments were performed in a dimly lit room. Each single neuron recording experiment consisted of two separate blocks. The baseline block comprised approximately 80 trials. The target moved at a constant speed of 20 deg/s for 850 ms in one of eight directions. Target motion followed a standard step-ramp trajectory (*Rashbass, 1961*), with a 3° eccentric step to minimize the occurrence of saccades during the initiation of pursuit. A learning block comprised about 400 trials. Each trial delivered a sequence of target motions described in the 'Results'.

Monkeys were required to keep their eyes within a reward window that was ±1° during fixation, ±2° during smooth target motion, and ±4 or 5° for 400 ms after an instructive change in target direction. The monkey received a droplet of juice at the end of the trial if he had satisfied the fixation requirements throughout the trial. To avoid punishing the monkey for his inescapable response latencies in learning trials, fixation requirements were suspended for the 250 ms between the onset of target motion and the time of the instructive change in target motion. The magnitude of learned eye movements was small and did not affect the chances of reward in either a positive or a negative way.

## Data acquisition and analysis

To estimate horizontal and vertical eye position, we scaled the eye position signals from a magnetic search coil system. The signals were passed through an analog circuit that rejected signals at frequencies above 25 Hz (−20 dB per decade) and differentiated signals at lower frequencies to create voltages proportional to horizontal and vertical eye velocity. Analog signals related to eye position and velocity were sampled at 1 kHz on each channel and stored for off-line analysis with single-unit recordings.

To record the activity of single Purkinje cells in the floccular complex, glass-insulated platinum–iridium microelectrodes manufactured in our laboratory were introduced daily through the previously-implanted cylinder. Purkinje cells showed a high level of spontaneous simple-spike firing and occasional complex-spike (CS) responses. Purkinje cells in the floccular complex showed strong modulation of simple-spike firing rate when the monkey tracked sinusoidal target motion along the horizontal or vertical axis. Extracellular action potentials were amplified conventionally, filtered with a bandpass of 300 Hz to 3 kHz, digitized at 25 KHz, and stored in the computer. We used a software window discriminator to view the spike train for each trial in the dataset and identify simple-spikes and CS responses. The firing rate for simple-spikes was computed using a reciprocal interval algorithm (*Lisberger and Pavelko, 1986*). The CS responses were accumulated for each learning trial in bins with widths of 100 ms, and then were converted to the probability of a CS in each bin.

We analyzed the responses of 108 Purkinje cells whose CS responses remained isolated through the full learning protocol. The CS responses occur in floccular Purkinje cells in response to instructive target motions that are in the 'off'-direction for simple-spike firing during pursuit (*Stone and Lisberger, 1990*; *Medina and Lisberger, 2008*). Our analysis is based on the large majority of the Purkinje cells (group 1, 90/108) that showed CS probabilities more than three times their spontaneous level of 0.09 ± 0.02 (mean ± SD). We excluded from analysis a much smaller subpopulation (group 2, 18/108), whose CS probabilities after an instruction were less than 0.2 and never exceeded three times their spontaneous level.

Specific data analyses are explained at the relevant locations in the 'Results', and statistical tests are described either in the 'Results' or in the relevant figure legends.

## Computer simulations

We created a cerebellar circuit model with 1000 Purkinje cells and 100 neurons in the inferior olive. In keeping with the known anatomy of the cerebellum, each neuron in the inferior olive provided a climbing fiber for 10 Purkinje cells (*Sugihara et al., 2004* visualized at least 5). The performance of the model was nearly the same if we set the climbing fiber divergence ratio to be 5 or even 1. We computed the firing rate for each Purkinje cell as:

$$SS_{i,j} = (1 - R_{nn})\alpha_{i,j} + R_{nn}\beta_j \tag{1}$$

where $SS_{i,j}$ is the firing rate of the $i^{th}$ Purkinje cell on the $j^{th}$ trial, $R_{nn}$ is the trial-by-trial correlation of simple-spike firing rate between pairs of Purkinje cells; $\alpha_{i,j}$ is drawn for each Purkinje cell and trial from a normal distribution with a mean of 100 and a standard deviation of 18; and $\beta_j$ is drawn for each trial from the same random distribution but used for all Purkinje cells on that trial. We fixed the mean and standard deviation of the distribution of simple-spike firing to match the means in our data. We set $R_{nn}$ to be 0.3, which yielded neuron–neuron correlations that averaged 0.16.

For each model inferior olive neuron, we enforced the probability of a CS to be zero when the instruction was in the on-direction for simple-spike firing, and we biased the probability of a CS on the basis of the average simple-spike firing rate of a subset of the model Purkinje cells when the instruction was in the off-direction for simple-spike firing. First, we computed the (disynaptic) simple-spike effect on each model inferior olive neuron as:

$$IN_{k,j} = \frac{\displaystyle\sum_{i=kN}^{(k+1)N-1} SS_{i,j}}{N} \tag{2}$$

where $IN_{k,j}$ signifies the averaged input to the $k^{th}$ inferior olive neuron on the $j^{th}$ trial, and $N$ is the number of Purkinje cells' simple-spike firing rates that are averaged together to modulate the response

of each model inferior olive neuron. Then, we used a sigmoid curve to relate the probability that each inferior olive neuron would respond to an off-direction instruction:

$$P_{k,j}(IO) = 0.1 + \frac{0.5}{(1 + e^{-0.3(IN_{k,j}-100)})} \qquad (3)$$

where $P_{k,j}(IO)$ signifies the probability that the $k^{th}$ model inferior olive neuron will emit a response on the $j^{th}$ trial. We then determined whether a CS would occur on the $j^{th}$ trial in the $i^{th}$ model Purkinje cell according to the Boolean variable:

$$CS_{i,j} = (R_{CS} \cdot \delta_{k,j}) < P_{k,j}(IO) \qquad (4)$$

where $R_{CS}$ is a random number normally distributed around 1 (standard deviation: 0.4) that determines the degree of synchrony across responses of model inferior olive neurons; and $\delta_{k,j}$ is a random variable with a value between 0 and 1 that is used to convert the probability of a response in the $k^{th}$ inferior olive neuron on the $j^{th}$ trial into an all or none response. To implement the 10-to-1 relationship between inferior olive neurons and Purkinje cells, *CS* and *P(IO)* use indexes of *i* vs *k*, according to the relationship: *k=ceil((i-1)/10)*. We changed the synchrony across CS responses in different model neurons by changing the range used to draw values of $R_{CS}$, and we eliminated synchrony across inferior olive outputs by setting the value of $R_{CS}$ equal to one.

We ran the model as a series of trials, just as we had done with our data, with a random stream of on- and off-direction instructions. For each trial, we calculated the weakly correlated firing rates of the population of model Purkinje cells (*Equation 1*), and subtracted 5 spikes/s to implement plasticity if a CS had occurred for that Purkinje cell on the prior trial. Next, we computed the CS responses on the current trial for each model Purkinje cell (*Equations 2–4*). CS-caused plasticity decayed by 2.5 spikes/s on each of the next two trials. The simple-spike firing rate for a given Purkinje cell was represented as a scalar. Thus, the results of analysis of the model outputs also were scalars that were meant to represent the mean simple-spike firing rate across the analysis interval in our data. To create figures that were easy to compare with the actual data, we used those scalars as gain factors for time-varying traces from the data. There did not seem to be any reason to simulate either neural or behavioral responses on a millisecond time case within each trial, so time is represented in our model only in terms of the progression from trial-to-trial in a learning block.

## Acknowledgements

We thank Megan Carey, Court Hull, and members of our laboratory for helpful comments on an earlier version of the manuscript and for helpful discussions. K MacLeod, E Montgomery, S Tokiyama, S Ruffner, D Kleinhesselink, D Wolfgang-Kimball, D Floyd, S Happel and K McGary provided invaluable technical assistance. Research supported by the Howard Hughes Medical Institute Research and by the National Eye Institute. The content is solely the responsibility of the authors and does not necessarily represent the official views of the NIH.

## Additional information

### Funding

| Funder | Grant reference number | Author |
| --- | --- | --- |
| Howard Hughes Medical Institute | Investigator | Stephen G Lisberger |
| National Institutes of Health | R01 EY003878 | Stephen G Lisberger |

The funders had no role in study design, data collection and interpretation, or the decision to submit the work for publication.

### Author contributions

YY, Conception and design, Acquisition of data, Analysis and interpretation of data, Drafting or revising the article; SGL, Conception and design, Analysis and interpretation of data, Drafting or revising the article

## Ethics

Animal experimentation: The Institutional Animal Care and Use Committees at UCSF (protocol AN08-36-25) and Duke (protocol A182-12-06) had approved all procedures used in this study in advance. Procedures were in accordance with the National Institutes of Health Guide for the Care and Use of Laboratory Animals. All surgery was performed under isofluorane anesthesia and analgesics were used to avoid pain or distress.

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
