## [Decision Letter]

Thank you for sending your work entitled “Cellular and circuit mechanisms that operate during the acquisition of cerebellum-dependent motor learning” for consideration at *eLife*. Your article has been favorably evaluated by a Senior editor and 3 reviewers, one of whom is a member of our Board of Reviewing Editors.

The Reviewing editor and the other reviewers discussed their comments before we reached this decision, and the Reviewing editor has assembled the following comments to help you prepare a revised submission.

In the present paper, Yang and Lisberger characterize climbing fiber-dependent one-trial signaling mechanisms and consequences for learning in smooth pursuit eye movements of macaques. This is an interesting manuscript with high-quality experiments that track the changes in simple and complex spike firing in Purkinje cells during learning. The data reveal that visual “instructive signals” that drive complex spikes (because of a deviation in the trajectory of a target) change simple spike rates on the next trial, and that simple spike and complex spike rates during that learned trial vary directly, consistent with disynaptic disinhibition between Purkinje and inferior olive cells.

The reviewers found the work well done and carefully analyzed. However, there was consensus that, in its present form, the manuscript is very hard to read. First, the task and analyses are complicated, and it is difficult to keep track of the time and trial windows when measurements are made. As written it is nearly impossible to follow the text and figures without multiple readings and diagram drawings in the margins to keep hold of the line of the arguments. The schematic in Figure 8 is quite helpful, but it comes too late. Thus, substantial rewriting of the text and an earlier flow-chart-like diagram (perhaps right in Figure 1, which has limited new information) might be useful to readers. Second, since this is one in a line of studies on a common topic, the authors sometimes slip into a jargon or shorthand that is difficult to follow, especially regarding “the plasticity” or references to parts of previous manuscripts. It would be informative if they spelled out a little more explicitly what is meant.

A major problem of this study is that the authors do not distinguish accurately enough between different types of modulation and learning. In the Introduction, they refer to multiple papers that characterized plasticity/learning in cerebellar slices and correctly point out that it remains to be shown how these forms of plasticity mediate learning in the intact animal. Subsequently, the authors show their data on single trial and trial-over-trial learning as observed in vivo. The crucial difference is that trial-over-trial learning is a form of short-term modulation that affects Purkinje cell spike firing, but it does not seem to be involved in any form of long-term motor learning, as implied in the Abstract. It is important that the authors either 1) re-write the manuscript to avoid over-selling the presented mechanism as being related to cerebellum-dependent motor learning (which most readers would assume means long-term learning), or 2) provide a model/explanation how trial-over-trial learning can indeed mediate more than a short-term modulation of spike firing.

An additional major concern is that the authors refer excessively to “mechanism,” even though this paper doesn't fit into the usual categories of long-term learning and the desired correlation with some kind of synaptic mechanisms. But it actually does something more important, by showing how Purkinje and IO firing fluctuates on a time scale that is too long for well-described kinds of short-term synaptic plasticity, but not necessarily linked to well-described long-term synaptic plasticity. In this way, the work goes beyond those categorizations and is what we need to clear up the rather trivial categorizations that have often stymied the field. But the authors do themselves no favors by their oblique invocations of synaptic plasticity mechanisms.

All these concerns could be addressable with a major re-writing of the text.

Specific comments:

1) The results are discussed in the wrong context, i.e., the context of long-term learning. An inexperienced reader might get the impression that a novel learning mechanism is presented, but it really is not. The authors should present the data in the context of short-to medium-term circuit modulations. They could speculate on how this relates to long-term plasticity and learning. As written, the Discussion implies that previous descriptions of potential learning mechanisms are incomplete, and that here a novel mechanism is presented that finally solves the issue. Since this is clearly not the case, major re-writing is necessary.

2) Abstract: “A single complex-spike response driven by an instruction for learning causes plasticity in the strength of a Purkinje cell's mossy fiber inputs.” This is an example of the sort of shorthand that makes the paper difficult. As the authors certainly know, Purkinje cells don't have mossy fiber inputs, but granule cell inputs (which have mossy fiber inputs). Perhaps more importantly, the only plasticity shown here is in the firing rates of Purkinje cells, not in strengths of any inputs. In fact, this idea should be stated more explicitly throughout. The authors very reasonably and even-handedly offer these results as a body of work to drive future investigation, which presumably will include cellular level studies. It is true that a good guess is that parallel fiber inputs may change but mossy fiber-granule synapses, or inhibitory synapses may play significant roles. But these data don't address that synaptic question. See also comment 6.

3) Description of the analysis that goes into Figure 3: “Next we measured the strength of the relationship....” This explanation of the sequence of analyses is understandable, but the rationale for doing this is not altogether clear. A little more explanation of why this was done might help the reader gain an intuitive grasp of what the predicted underlying factors are.

4) The authors' interpretation of why the on-direction and off-direction gave such different outcomes is unclear.

5) Point 4 of the list in the “Neural mechanisms of trial-over-trial depression and learning” section: the model figure includes granule cells and DCN cells but they are not described in the list. Are they actually modeled, or are only the Purkinje and IO modeled and the role of the DCN and granule cells is implicit? If it is the latter, it would be very helpful to indicate that the idea is that when the Purkinje rate goes high, the expectation is that the IO rate will also go high because of disynaptic disinhibition. Are the granule cells in the model doing anything?

6) Discussion: The opening to the Discussion stresses “cellular and circuit mechanisms” and the word “mechanisms” is used 6 times in the first paragraph. This is not really appropriate, since the strength of the work is that data and model describe changes in spiking that occur during learning. But there is really nothing about mechanism. We see what happens but these experiments do not probe how it happens. This is not a shortcoming. It poses a challenge to those who study synaptic mechanisms to account for how the real brain does what it does. But it does not make sense to say that this paper is about mechanisms. The word is used repeatedly and inappropriately for the next few paragraphs and should be removed throughout.

---

## [Author Response]

*The reviewers found the work well done and carefully analyzed. However, there was consensus that, in its present form, the manuscript is very hard to read. First, the task and analyses are complicated, and it is difficult to keep track of the time and trial windows when measurements are made. As written it is nearly impossible to follow the text and figures without multiple readings and diagram drawings in the margins to keep hold of the line of the arguments. The schematic in*
Figure 8
*is quite helpful, but it comes too late. Thus, substantial rewriting of the text and an earlier flow-chart-like diagram (perhaps right in*
Figure 1*, which has limited new information) might be useful to readers. Second, since this is one in a line of studies on a common topic, the authors sometimes slip into a jargon or shorthand that is difficult to follow, especially regarding “the plasticity” or references to parts of previous manuscripts. It would be informative if they spelled out a little more explicitly what is meant*.

We agree that our paper reports complicated experiments and we have taken a number of steps to help our readers. First, we have changed the x-axes in Figures 2, 3, 4 and 5 so that time zero is the time of the onset of the learning instruction. Second, we have made some minor changes so that the analysis intervals are consistent throughout the paper. We have added shading to all relevant graphs to show the analysis intervals, with simple-spike analysis intervals in light pink and complex-spike analysis intervals in light gray. Third, we have added a brief section early in the Results that is intended to orient the reader to the analysis intervals and to provide a few hints on reading the graphs. Fourth, we have highlighted the two important features of cerebellar function revealed by our paper as shading in Figure 1, and we have changed the discussion of Figure 1 to foreshadow where we are going. Fifth, we have looked critically for and removed jargon. As we revised the paper, we have identified a number of places where revisions in the text should make the paper clearer.

We considered moving Figure 8 to the front of the paper, but we decided that it should remain at the end of the paper because it depends on concepts that are demonstrated in the Results. Also, we think that most readers will benefit from the orientation to the cerebellar circuit that is provided by Figure 1. To follow the spirit of the reviewers’ suggestions, we have tried to use Figure 1 to better advantage in introducing our study.

*A major problem of this study is that the authors do not distinguish accurately enough between different types of modulation and learning. In the Introduction, they refer to multiple papers that characterized plasticity/learning in cerebellar slices and correctly point out that it remains to be shown how these forms of plasticity mediate learning in the intact animal. Subsequently, the authors show their data on single trial and trial-over-trial learning as observed in vivo. The crucial difference is that trial-over-trial learning is a form of short-term modulation that affects Purkinje cell spike firing, but it does not seem to be involved in any form of long-term motor learning, as implied in the Abstract. It is important that the authors either 1) re-write the manuscript to avoid over-selling the presented mechanism as being related to cerebellum-dependent motor learning (which most readers would assume means long-term learning), or 2) provide a model/explanation how trial-over-trial learning can indeed mediate more than a short-term modulation of spike firing*.

We agree with the reviewers. In the final analysis, we think that the findings in this paper will prove to have important implications for the mechanisms of long-term learning. However, the data that support our thinking will appear in another paper. We have rewritten the paper to be clear that we are studying very short-term effects that we call “trial-over-trial learning”.

*An additional major concern is that the authors refer excessively to “mechanism,” even though this paper doesn't fit into the usual categories of long-term learning and the desired correlation with some kind of synaptic mechanisms. But it actually does something more important, by showing how Purkinje and IO firing fluctuates on a time scale that is too long for well-described kinds of short-term synaptic plasticity, but not necessarily linked to well-described long-term synaptic plasticity. In this way, the work goes beyond those categorizations and is what we need to clear up the rather trivial categorizations that have often stymied the field. But the authors do themselves no favors by their oblique invocations of synaptic plasticity mechanisms*.

We agree with the editors and reviewers, and we appreciate the help this comment has provided us in putting our data in an appropriate context. We have rewritten the entire paper to incorporate the point-of-view they have suggested. “Mechanism” now appears in our text only twice, and we have referred to the phenomenon we study as “plasticity of simple-spike firing rate”.

*Specific comments*:

*1) The results are discussed in the wrong context, i.e., the context of long-term learning. An inexperienced reader might get the impression that a novel learning mechanism is presented, but it really is not. The authors should present the data in the context of short-to medium-term circuit modulations. They could speculate on how this relates to long-term plasticity and learning. As written, the Discussion implies that previous descriptions of potential learning mechanisms are incomplete, and that here a novel mechanism is presented that finally solves the issue. Since this is clearly not the case, major re-writing is necessary*.

See response to comment 2, below.

*2) Abstract: “A single complex-spike response driven by an instruction for learning causes plasticity in the strength of a Purkinje cell's mossy fiber inputs.” This is an example of the sort of shorthand that makes the paper difficult. As the authors certainly know, Purkinje cells don't have mossy fiber inputs, but granule cell inputs (which have mossy fiber inputs). Perhaps more importantly, the only plasticity shown here is in the firing rates of Purkinje cells, not in strengths of any inputs. In fact, this idea should be stated more explicitly throughout. The authors very reasonably and even-handedly offer these results as a body of work to drive future investigation, which presumably will include cellular level studies. It is true that a good guess is that parallel fiber inputs may change but mossy fiber-granule synapses, or inhibitory synapses may play significant roles. But these data don't address that synaptic question. See also comment 6*.

We found comments 1 and 2 particularly apt, and we have tried to restructure our Abstract, Introduction, and Discussion in terms of “plasticity of simple-spike firing”. We mentioned in the Abstract and Introduction that the short-term phenomena we have studied might relate to long-term plasticity and learning. Otherwise, we stick closely to the actual time frame of the measurements presented here.

*3) Description of the analysis that goes into*
Figure 3*: “Next we measured the strength of the relationship....” This explanation of the sequence of analyses is understandable, but the rationale for doing this is not altogether clear. A little more explanation of why this was done might help the reader gain an intuitive grasp of what the predicted underlying factors are*.

We have provided some introductory material to motivate this analysis.

*4) The authors' interpretation of why the on-direction and off-direction gave such different outcomes is unclear*.

We do not understand why the relationship between complex-spike probability and simple-spike firing rate is different for on- versus off-direction data. We have revised this paragraph to give a little more information and we have added a brief statement a) explaining what this result might mean and b) confessing that we don’t have a definitive interpretation.

*5) Point 4 of the list in the “Neural mechanisms of trial-over-trial depression and learning” section: The model figure includes granule cells and DCN cells but they are not described in the list. Are they actually modeled, or are only the Purkinje and IO modeled and the role of the DCN and granule cells is implicit? If it is the latter, it would be very helpful to indicate that the idea is that when the Purkinje rate goes high, the expectation is that the IO rate will also go high because of disynaptic disinhibition. Are the granule cells in the model doing anything*?

Our schematic diagram is “anatomically correct”, to a first approximation. But, the reviewers are correct that we have modeled only the Purkinje cells and the inferior olive. We have resolved this by 1) removing the granule cell from the diagram in Figure 5 and 2) clarifying how disynaptic disinhibition would work in the circuit and our model. We also have added some highlighting to the figures to emphasize the two key circuit mechanisms used by our model.

*6) Discussion: The opening to the Discussion stresses “cellular and circuit mechanisms” and the word “mechanisms” is used 6 times in the first paragraph. This is not really appropriate, since the strength of the work is that data and model describe changes in spiking that occur during learning. But there is really nothing about mechanism. We see what happens but these experiments do not probe how it happens. This is not a shortcoming. It poses a challenge to those who study synaptic mechanisms to account for how the real brain does what it does. But it does not make sense to say that this paper is about mechanisms. The word is used repeatedly and inappropriately for the next few paragraphs and should be removed throughout*.

We have largely removed the work “mechanisms” from the vocabulary of the paper and have been more careful and circumspect in our writing.